# Learning Robust Social Strategies with Large Language Models

## Abstract

As agentic AI becomes more widespread, agents with distinct and possibly conflicting goals will interact in complex ways. These multi-agent interactions pose a fundamental challenge, particularly in social dilemmas, where agents' individual incentives can undermine collective welfare. While reinforcement learning (RL) has been effective for aligning large language models (LLMs) in the single-agent regime, prior small-network results suggest that standard RL in multi-agent settings often converges to defecting, self-interested policies. We show the same effect in LLMs: despite cooperative priors, RL-trained LLM agents develop opportunistic behavior that can exploit even advanced closed-source models. To address this tendency of RL to converge to poor equilibria, we adapt a recent opponent-learning awareness algorithm, Advantage Alignment, to fine-tune LLMs toward multi-agent cooperation and non-exploitability. We then introduce a group-relative baseline that simplifies advantage computation in iterated games, enabling multi-agent training at LLM scale. We also contribute a novel social dilemma environment, *Trust-and-Split*, which requires natural language communication to achieve high collective welfare. Across a wide range of social dilemmas, policies learned with Advantage Alignment achieve higher collective payoffs while remaining robust against exploitation by greedy agents.

## 1 Introduction

LLMs undergo large-scale pretraining, instruction tuning, and reinforcement learning, and continue to exhibit increasingly advanced capabilities (Guo et al., 2025). Coupled with decreasing deployment costs and improved adaptability to downstream tasks, these trends enhance the commercial and practical viability of LLM agents across a wide range of applications. Recent efforts are already translating this potential into concrete systems. Anthropic's Model Context Protocol (MCP; Anthropic, 2024) enables an LLM to interact with external systems and become more capable as an autonomous decision-making agent. At the same time, LLM agents are now being deployed in real applications, from code generation and software development assistance (Chen et al., 2021) to e-commerce transactions and personalized information curation (OpenAI, 2024a;b). New infrastructure is also emerging to support agent-agent interaction, such as Google's Agent2Agent protocol (Agent2AgentProtocol, 2024), which enables collaboration among LLM-based agents with varying capabilities, potentially across different organizations.

Despite rapid progress, LLM behavior in multi-agent settings remains poorly understood. One common scenario involves agents with conflicting goals that discourage cooperation, even when cooperation would lead to better outcomes for all. These situations, known as *social dilemmas* (Rapoport & Chammah, 1965), frequently arise in real-world contexts where agents face a tension between individual gain and collective welfare. They appear in everyday scenarios such as navigating traffic, as well as in more complex settings such as business negotiations or international policy coordination. A recent example is the case of many LLM crawlers downloading training data from small code-hosting websites, causing them to be overwhelmed with DDoS-like traffic (SourceHut, 2025). Such interactions are analogous to the famous *tragedy of the commons*, a social dilemma concerning the maintenance of public goods, where self-interested behavior leads to resource depletion. Such cases illustrate the types of social dilemmas that may arise in complex environments where LLMs are increasingly expected to act and interact autonomously.

Learning to resolve these scenarios is typically framed within multi-agent reinforcement learning (MARL). Social dilemmas are a specific subclass of MARL problems that are mixed-motive; neither fully cooperative nor fully competitive. Unlike single-agent RL, where an agent improves an objective in a static environment, in MARL each agent must adapt to the strategies of other agents, who can also be learning over time. This leads to non-stationarity, since the policy of each learning agent affects the collective outcome. Initial attempts using MARL to play social dilemmas were unsuccessful. Training agents based on small neural networks with naive MARL resulted in sub-optimal greedy strategies (Sandholm & Crites, 1996). To address this, Foerster et al. (2018) introduced Opponent Shaping (OS), an RL paradigm that explicitly considers agent interactions in hopes of steering their dynamics toward mutually beneficial outcomes. LOLA, the first OS algorithm, is capable of finding the pareto-optimal strategy of *tit-for-tat* in simple social dilemmas like the Iterated Prisoner's Dilemma.

Prior work largely focused on teaching such tabula-rasa agents reciprocity—punishing greed and rewarding cooperation—where the central obstacle was that uninformed policies gravitated toward short-sighted, self-interested strategies. By contrast, LLMs arrive with rich priors and human-like social norms induced by pretraining and post-training (instruction tuning/RLHF) (Ross et al., 2024), potentially altering the learning dynamics and failure modes in multi-agent settings. This raises a key question: when fine-tuned with naive MARL, do LLMs have the same failure modes as small networks, or do their human-biased priors mitigate them? Since LLM agents already interact in the wild, understanding this behavior is an important research challenge.

To study this behavior, we introduce a novel testbed for social dilemmas in the LLM setting. The testbed includes small-scale social dilemma environments (Duque et al., 2025a) which we extend to the textual domain, as well as our new communication-based Trust-and-Split environment, designed to measure both cooperation and non-exploitability. Using this testbed, we conduct extensive experiments across a range of modern LLMs and find that naive MARL consistently produces greedy behavior across all environments. Probing further, we show that even state-of-the-art closed-source models are exploitable when facing agents trained with naive MARL. These results underscore that current LLMs are not yet prepared to operate robustly in real-world multi-agent settings. Together, they provide a novel insight: failure in multi-agent settings can arise simply from naive MARL fine-tuning in a social dilemma.

To overcome this issue, we adapt Advantage Alignment (Duque et al., 2025b), a recent opponent shaping algorithm, to train LLM agents that cooperate reliably and resist exploitation in social dilemma environments. Specifically, we introduce a group-relative baseline to compute advantages in multi-round settings and implement an agent buffer with LoRA (Hu et al., 2022) to maintain diversity during training. When trained with Advantage Alignment using these design choices, we find that agents learn the non-exploitable and effective *tit-for-tat* strategy in the classic Iterated Prisoner's Dilemma. In complex environments like Split No-Comm and Trust-and-Split, Advantage Alignment agents learn to cooperate with cooperative players as well as themselves, while remaining robust against greedy players.

In summary, our key contributions are:

- Developing a social dilemma testbed for LLMs, including standard environments and our novel Trust-and-Split environment, where achieving high welfare requires communication.
- Demonstrating that naive MARL leads to greedy, suboptimal agents across this testbed for a range of open-source LLMs, and that even state-of-the-art closed-source LLMs are vulnerable to exploitation by greedy RL-trained agents.
- Adapting the Advantage Alignment algorithm (Duque et al., 2025a) to the LLM setting to train agents that achieve cooperative, non-exploitable behavior across this testbed.

## 2 BACKGROUND

### 2.1 MARKOV GAMES

An $n$-agent Markov game (Shapley, 1953) is defined as a tuple $(\Pi, \mathcal{S}, \mathcal{A}, \mathcal{R}, \mathcal{P}, \gamma)$. $\mathcal{S}$ is a set of possible states. $\mathcal{A}$ is a set of functions $\mathcal{A}^1, \ldots, \mathcal{A}^n$ where $\mathcal{A}^j(S)$ gives the set of possible actions of agent $j$ at state $S$. $\mathcal{R}$ is the set of reward functions $\{r^1, \ldots, r^n\}$ where $r^j : \mathcal{S} \times \mathcal{A} \to \mathbb{R}$

is the reward function of agent $j$. $\mathcal{P}$ is the transition function that assigns a probability distribution to each transition $\mathcal{P}(S \times \mathcal{A} \rightarrow S')$. $\Pi$ is the set of policies $\{\pi^1, \ldots, \pi^n\}$, each $\pi^j$ mapping any state $S$ to a probability distribution over $\mathcal{A}^j(S)$. $\gamma$ is the discount factor on the returns. The expected discounted return of player $j$ is $J^j(\Pi) = \mathbb{E}_{\tau \sim \mathrm{Pr}_\mu^\Pi}\left[\sum_{t=0}^{\infty} \gamma^t r^j(s_t, \mathbf{a}_t)\right]$, where $\mathrm{Pr}_\mu^\Pi$ is the distribution of trajectories induced by the initial state distribution $\mu$ and the set of policies $\Pi$, $\mathbf{a}_t$ is the set of actions at time $t$. The probability of a trajectory $\tau$ under distribution $\mathrm{Pr}_\mu^\Pi$ is

$$\mu(s_0) \prod_{t=1}^{\infty} \left[ \mathcal{P}(s_t | s_{t-1}, a_{t-1}^1, \ldots, a_{t-1}^n) \prod_{j=1}^{n} \pi^j(a_{t-1}^j | s_{t-1}) \right].$$

## 2.2 MULTI-AGENT REINFORCEMENT LEARNING

In a Markov Game, each agent $j$ attempts to maximize its objective. For each agent, the multi-agent state-value function is defined as $V^j(s) := \mathbb{E}_{\mathbf{a} \sim \Pi(s)}\left[r^j(s, \mathbf{a}) + \gamma \mathbb{E}_{s' \sim \mathcal{P}(s,\mathbf{a})}\left[V^j(s')\right]\right]$, the action-value function as $Q^j(s, \mathbf{a}) := r^j(s, \mathbf{a}) + \gamma \mathbb{E}_{s' \sim \mathcal{P}(s,\mathbf{a})}\left[V^j(s')\right]$, and the advantage function as $A^j(s, \mathbf{a}) := Q^j(s, \mathbf{a}) - V^j(s)$. For any Markov Decision Process, the REINFORCE (Williams, 1992) algorithm uses unbiased estimates of the gradient of the state-value function with respect to the parameters of $\pi$ in order to perform gradient ascent. GRPO (Shao et al., 2024) reduces the variance of REINFORCE by introducing a simple baseline subtraction. GRPO can easily be extended to the multi-agent case by independently updating each policy $j$ with $\nabla_{\theta^j} J^j(\Pi) = \mathbb{E}_{\tau \sim \mathrm{Pr}_\mu^\Pi}\left[\sum_{t=0}^{\infty} \gamma^t A^j(s_t, \mathbf{a}_t) \nabla_{\theta^j} \log \pi^j(\mathbf{a}_t^j | s_t)\right]$, the multi-agent advantage function being computed using a GRPO-style baseline (described in section 3). In the context of this paper, the naive MARL algorithm follows this formulation and is called *multi-agent GRPO*. We also consider the naive cooperative variant *multi-agent GRPO with sum of rewards*, which is algorithmically equivalent except for the fact that the reward functions of each agent are changed to $r(s, \mathbf{a}) := \sum_{j=1}^{n} r^j(s, \mathbf{a})$. That is, each agent optimizes for the sum of expected discounted returns across all agents. This formulation encourages agents to learn policies that maximize overall welfare rather than focusing on individual benefits.

## 2.3 SOCIAL DILEMMAS

In a zero-sum game, the agents' payoffs always add up to zero; every gain for one side is matched by an equal loss for the other. Consequently, in a two-player zero-sum setting, cooperation does not offer benefit. In this work, we focus on general-sum games, where total payoffs are not fixed, and agents may improve their outcomes without necessarily diminishing those of others, thereby creating the possibility of mutually beneficial cooperation. More precisely, we focus on social dilemmas, general-sum games in which agents face a tension between their short-term individual benefit and long-term collective welfare. In these settings, each agent has a short-term incentive to act selfishly (i.e., not cooperate), but if all agents do so, the resulting outcome leads to reduced overall welfare, i.e. a lower total sum of discounted returns *for all agents*. However, if an agent is unconditionally cooperative, other rational agents will exploit it and reduce its welfare to increase theirs. The focus of this paper is on a stronger alternative strategy, which *incentivizes* rational agents to behave in its best interest, achieving high collective welfare while avoiding exploitation.

## 2.4 OPPONENT SHAPING

Prior work shows that small neural networks trained with naive MARL tend to converge to the *Always Defect* strategy in IPD (Sandholm & Crites, 1996). More recently, Foerster et al. (2018) demonstrated that this undesirable outcome also arises with policy gradient methods. These approaches assume that the environment is stationary, which is valid in a single-agent setting, but not in a multi-agent setting where other learning agents create non-stationarity. LOLA (Foerster et al., 2018) removed the assumption of a static environment in markov games and included a model of a learning agent in its update. By explicitly modeling how opponent learning is affected by an agent's action, LOLA was able to learn the *tit-for-tat* strategy in IPD. Unfortunately, LOLA's computational complexity is quadratic in the number of parameters of the agent, making it impractical for LLMs.

Advantage Alignment (Duque et al., 2025a) is an opponent-shaping algorithm that instead focuses on the Q-values of both the agent and its opponent. Assuming that agents act proportionally to the

exponent of their Q-value, Advantage Alignment aims to align an opponent's Q-value with its own. This leads to a simple modification to the advantages used in the policy gradient term of a REIN-FORCE estimator. Advantage Alignment has been shown to solve social dilemmas in scenarios with high dimensional state representations (e.g. pixel spaces), partial observability, and continuous action spaces. Given its performance in complex scenarios, we chose Advantage Alignment as a prime candidate to train LLMs to find cooperative and non-exploitable strategies.

## 3 ADVANTAGE ALIGNMENT FOR LLMS

Advantage Alignment algorithms (Duque et al., 2025b) extend the regular policy gradient update with a reweighting of the action gradients that includes the agent's past advantages and the advantage of its opponent. For a pair of policies, the update for $\theta^1$ is

$$\mathbb{E}_{\tau \sim \mathrm{Pr}_\mu^{\pi^1, \pi^2}} \left[ \sum_{t=0}^{\infty} \gamma^t \left( A_t^1 + A_t^2 \beta \gamma \sum_{k<t} \gamma^{t-k} A_k^1 \right) \nabla_{\theta^1} \log \pi^1(a_t|s_t) \right] \tag{1}$$

where $A_x^j$ is shorthand for $A^j(s_x, a_x, b_x)$. The update is symmetric for $\theta_2$.

Estimating advantages with value networks has proven challenging in the context of LLM training, often leading to unstable or ineffective results (Kazemnejad et al., 2024). Recent work such as RLOO (Ahmadian et al., 2024) and GRPO (Shao et al., 2024) has shown that baseline-based approaches provide more stable and efficient advantage estimates. These approaches sample multiple trajectories for a given prefix, and compute the advantage for each trajectory as the difference between its discounted return and the mean discounted return of the remaining trajectories. However, scaling this approach to multi-round, multi-agent settings is infeasible because the number of trajectories needed grows exponentially. In our experiments, we build on these ideas and extend them to multi-agent LLM training. We divide each batch of rollouts into $k$ common random number (CRN) groups, each of which uses a fixed random seed to generate the environment stochasticity. This ensures that, within a CRN group, the variance in discounted returns comes only from the agent's actions and not from the environment. This is similar in spirit to GRPO and RLOO, except trajectories share a fixed environment context rather than a shared prefix. In particular, let $A^i(s_t, a_t)$ denote the advantage for agent $i$. We estimate it using a leave-one-out group baseline computed over the $k$ games of its CRN group at each time step $t$: $G(a_t^{(i)}, s_t) - \frac{1}{k-1} \sum_{j \neq i} G(a_t^{(j)}, s_t)$ where $G(a_t^{(i)}, s_t)$ is the discounted return for action $a_t^{(i)}$ taken in state $s_t$. This group-relative baseline avoids the need for a learned value function, simplifies advantage computation, and enables multi-turn RL with LLMs in our multi-agent settings. We refer to this algorithm as multi-agent GRPO in the rest of the paper.

Each agent's policy $\pi_i$ is parameterized by $\theta_i$ and implemented via LoRA finetuning (Hu et al., 2022). Throughout our experiments, we refer to the first player as *Alice* and the second as *Bob*. We use self-play, i.e., the same set of parameters for both agents, conditioned on different game contexts based on their roles. This ensures that memory usage doesn't scale with the number of agents and the model size we used is sufficient to handle the complexity of the different roles. Maintaining opponent diversity is essential for self-play, and it is particularly important in social dilemmas, where defection equilibria can trap learning. Without diversity, exploration suffers and agents may remain stuck in defecting strategies. Following Duque et al. (2025a), we preserve opponent diversity through an agent buffer that stores earlier versions of the self-play agent. This is straightforward to implement because each agent is represented by a LoRA checkpoint, roughly $0.1\%$ of the model parameters, which can be saved and reloaded with minimal overhead. For each game, with probability $\rho$, the opponent is sampled from the agent buffer. With probability $1 - \rho$, the opponent is simply the current version of the agent using the latest LoRA parameters. We use $\rho = 1/2$ as the default setting, and it works well in our experiments. For both multi-agent GRPO and its sum-of-rewards variant, the agent buffer made no noticeable difference, and for computational reasons we did not apply it to these methods.

## 4 SOCIAL DILEMMA TESTBED

In this section, we study the behavior of LLM agents trained with naive MARL in social dilemma environments. To support this, we develop a novel testbed tailored for LLMs to evaluate the effects

Figure 1: One round of Trust-and-Split. Each player receives a private rock-paper-scissors hand that determines how much they value the coins, sends one message in turn, and then submits a proposal. Payoffs follow the split rule. Both hands and proposals are revealed before the next round starts.

of MARL training on cooperation and resistance to exploitation. An exact description of the game prompt in all the environments is provided in Appendix 13.

**Iterated Prisoner's Dilemma** IPD is a two-player game where agents repeatedly and simultaneously choose to either *Cooperate* (C) or *Defect* (D) in each round. The per-round pay-off matrix used in our experiments is provided in Table 12 in the appendix. We include IPD in our testbed because it is one of the most widely studied social dilemmas. However, since it is also likely presented in the training data of LLMs, we obfuscate the nature of the game by removing any mention of "Prisoner's Dilemma" and replace the action labels *Cooperate* and *Defect* with *A* and *B*, respectively. This allows us to test how well LLMs generalize beyond memorization and to examine how RL interacts with any prior knowledge the model may have about this social dilemma.

**Split No-Comm** This environment is a textual version of the negotiation game used in Duque et al. (2025b). In this game, there are three item categories (hats, books and balls) to split at each round. The values of each item are public for both agents. At each round, item values are sampled as follows: (1) each item category is assigned a value of either 1 or 10 at random, (2) at least one item category must have different values for the two agents, creating a conflict and a social dilemma, and (3) the total value across all items is the same for both agents in that round. Proposals and payoffs are revealed after the end of each round. This variant supports reciprocity without the need for communication. The split rule (proposal mechanism) from the Negotiation Game (Cao et al., 2018; Duque et al., 2025a), provides a better learning signal for training agents in this dilemma. More precisely, let $p_{k,a}$ be the proposal for the $k$'th item category from agent $a$ and $q_k$ be the quantity available. The allocation received by agent $a$ is $q_{k,a} = q_k p_{k,a} / \max(q_k, p_{k,a} + p_{k,b})$ and similarly for agent $b$. The resulting payoffs are $v_a \times q_{k,a}$ and $v_b \times q_{k,b}$ respectively. This particular choice removes the need for explicit agreement and ensures that both agents receive a learning signal every round.

**Trust-and-Split** While IPD and Split No-Comm capture the fundamental dilemma, they lack the richness of real-world strategic interactions. Existing negotiation environments involve longer interactions (Davidson et al., 2024; Lewis et al., 2017b), which make them less feasible to train and more difficult for characterizing robust strategies that maximize collective payoff. Moreover, Liao et al. (2024b) find that LLMs up to the scale of 70B struggle to follow instructions in multi-item settings across multiple rounds. To address these limitations, we propose Trust-and-Split, a novel environment that builds on Split No-Comm by adding communication. Trust-and-Split uses a single item, coins, which avoids the complexity of multi-item negotiation while still requiring communication for effective performance. A visualization of a round in this environment is detailed in Figure 1. At the beginning of each round, each player is assigned an exclusive private hand among {rock, paper, scissors}. The agent with the lower hand values each coin at 1, while the agent with the upper hand values each coin at 10. Since neither player knows the other's hand, they are incentivized to communicate to infer values and play effectively. Each agent can then negotiate with the other agent, one message at a time. We currently limit the number of messages to one per agent to ensure that we can train these agents across multiple rounds. The setup also allows for a variety of behaviors, including bluffing, exaggeration, and cooperative negotiation. After the messaging phase, both agents submit their proposals simultaneously. They then receive their payoffs based on their coin values and the quantities allocated by the split rule. Before continuing to the next round, the hands

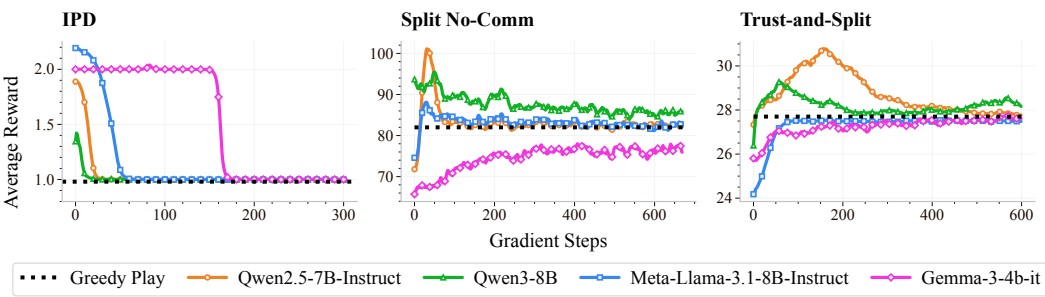

Figure 2: Training curves of multi-agent GRPO on several open-source LLMs across IPD, Split No-Comm, and Trust-and-Split. In all environments, average rewards converge to the greedy payoff levels, showing that naive MARL drives LLMs toward defecting strategies in social dilemmas.

and proposals are revealed to both players, allowing reciprocity. In this environment, the starting agent alternates every round, and hands are assigned so that in expectation, both agents receive an equal number of upper hands. The strategy that maximizes payoffs for both agents is to truthfully communicate hands and allocate all items to the agent who values them more in each round, while remaining non-exploitable.

## 5 EXPERIMENTS

Having introduced the testbed, we study how naive MARL interacts with LLMs in these settings and evaluate the effectiveness of Advantage Alignment. For games played over infinite rounds with a discount factor $\delta$, we found no empirical difference between training with fixed-length versus stochastic-length trajectories. For computational efficiency, we, therefore, use fixed-length trajectories throughout.

### 5.1 NAIVE MARL LEADS TO GREEDY BEHAVIOR WITH LLMS

In order to robustly demonstrate how MARL interacts with LLMs in social dilemmas, we train LLMs from several model families across all the environments. We use multi-agent GRPO with self-play as the learning algorithm and train only the LoRA parameters. Figure 2 shows that naive MARL consistently converges to greedy behavior across all environments and model families. In simpler environments like IPD, all models begin with higher than greedy average rewards but drift toward greedy play with training. In more complex environments such as Split No-Comm and Trust-and-Split, Qwen models briefly achieve higher average rewards than greedy play before collapsing back to greedy behavior, while Llama and Gemma models start with low performance and converge directly to greedy strategies. Qualitatively, in Split No-Comm, we find that agents learn to bid the highest for every item even when they value it less. In Trust-and-Split, agents communicate their private hands honestly but then propose to take all coins for themselves. These results show that naive MARL robustly leads to greedy behavior in social dilemma settings. Since LLM agents are likely to operate in scenarios that involve social dilemmas, this highlights the need for training methods that enable robust cooperation without being exploitable.

### 5.2 ADVANTAGE ALIGNMENT LEARNS ROBUST SOCIAL STRATEGIES

To address the shortcomings of naive MARL, we apply Advantage Alignment to learn robust policies in our environments. We run Advantage Alignment with eight different seeds across all environments and report average results in Figure 3.

For simpler environments such as IPD and Split No-Comm, the baseline agents, always-cooperate and always-defect agents can be hardcoded. In IPD, Always-Cooperate (Coop) agent always plays action *A*, equivalent to *Cooperate* and the Always-Defect (Defect) agent always plays action *B* equivalent to *Defect* as defined in section 2.3. In Split No-Comm, the Coop agent proposes 10 when its own value is 10 and the other player's value is 1, proposes 0 in the reverse case, and proposes 5 when both values are equal. The defect agent always proposes 10, regardless of the values. In

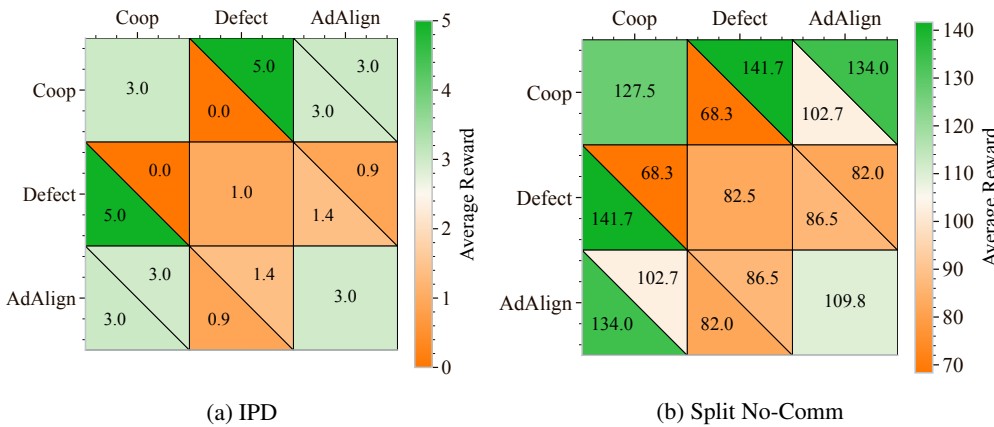

(a) IPD

(b) Split No-Comm

Figure 3: Average rewards when evaluating an Advantage Alignment (AdAlign) agent, an always-cooperate (Coop) agent, and an always-defect (Defect) agent. In IPD (left) and Split No-Comm (right), Advantage Alignment achieves near cooperative payoffs with itself and always-cooperate (Coop) while remaining robust against always-defect (Defect). Results are averaged over 8 seeds.

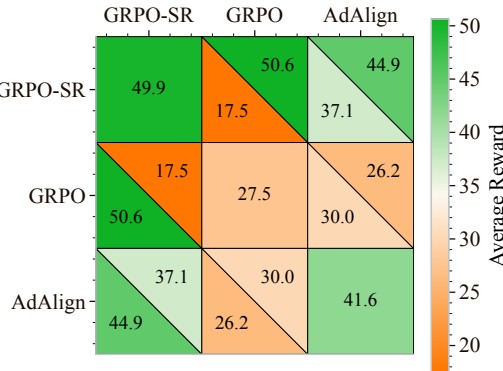

Figure 4: Average reward in Trust-and-Split when pitting an Advantage Alignment (AdAlign) agent against agents trained with multi-agent GRPO with sum of rewards i.e. Cooperators (GRPO-SR) and multi-agent GRPO i.e. Defectors (GRPO). Advantage Alignment cooperates with cooperative partners and itself, yet avoids being exploited by greedy agents. Results are averaged over 8 seeds.

the Iterated Prisoner's Dilemma, Advantage Alignment agents cooperate with themselves and with fully cooperative agents, while remaining robust against defection. The slight drop in performance against defectors comes from losing the first round, since the agent initially cooperates and receives a lower payoff on that round. In the Split No-Comm game, Advantage Alignment agents obtain about 86% of the full cooperation efficiency while still maintaining robustness. When paired with defectors, their performance decreases only slightly, indicating they are not easily exploitable. Qualitatively, we find that Advantage Alignment learns a *tit-for-tat* strategy in IPD, defecting when the other agent defects in the previous round and cooperating when the other agent cooperates (Appendix Figure 10). In Split No-Comm, it learns a strategy similar to *grim-trigger*, where a single defection can lead to persistent defection thereafter (Appendix Figure 11).

In Trust-and-Split, we cannot hardcode cooperative and defector policies because the environment requires communication. Instead, we train baseline agents using multi-agent GRPO, and its sum-of-rewards variant. As shown in Figure 4, multi-agent GRPO produces defectors that achieve low average reward when paired with themselves, while the sum-of-rewards variant produces cooperators that achieve the maximum possible reward with themselves. However, when these cooperators are paired with defectors, they are easily exploited, and the defectors obtain the maximum reward. Advantage Alignment agents learn to cooperate with cooperators and with themselves, achieving high average rewards. They learn to propose amounts close to 10 when holding higher hands (indicating

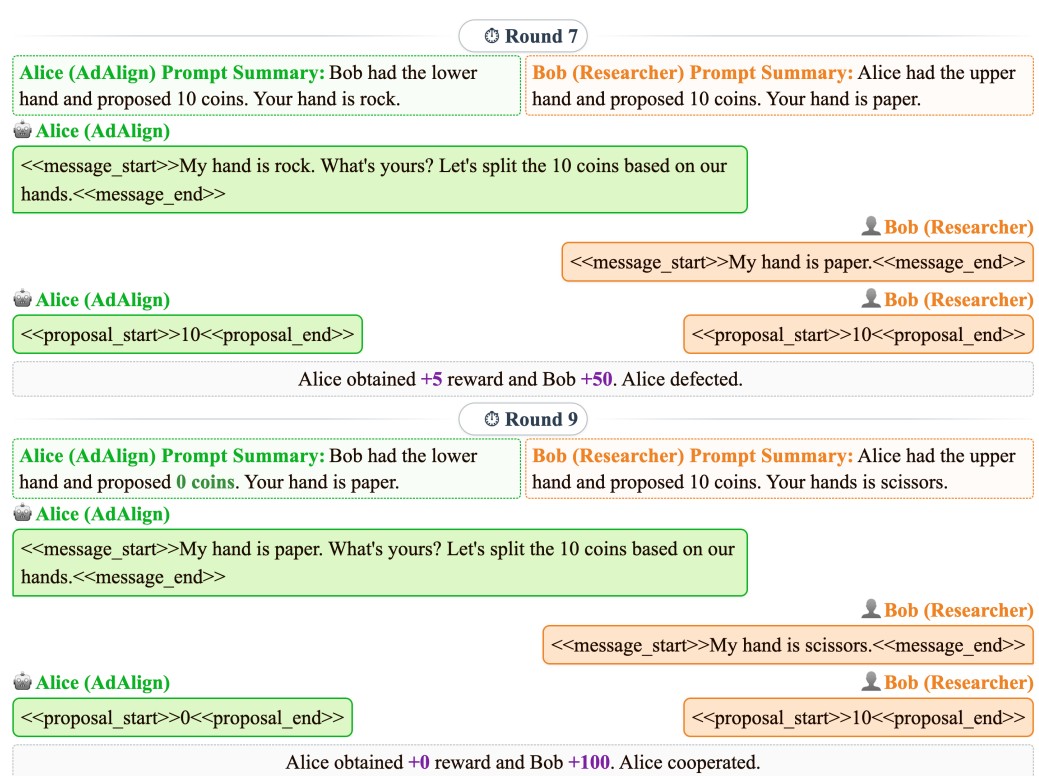

Figure 5: Example Trust-and-Split interaction showing the tit-for-tat behavior learned by Advantage Alignment. After *Bob* defects (as seen in the prompt summary of *Alice* for round 7), *Alice* defects in round 7, then returns to cooperation in round 9 once *Bob* cooperates again (shown in the prompt summary of *Alice* for round 9).

higher valuation) and amounts close to 0 when holding lower hands (indicating lower valuation), a strategy that maximizes collective payoffs as shown in Figure 7 in the appendix. At the same time, they remain non-exploitable and almost always defect when paired with defectors. We also find that Advantage Alignment agents are not brittle in the communication phase. They remain robust across different patterns of messages used to describe hands, as confirmed through qualitative interactions with the trained agents. Figure 5 illustrates the *tit-for-tat* behavior learned by Advantage Alignment in Trust-and-Split. At the start of round 7, *Bob* defected in the previous round by proposing 10 coins despite valuing them less. In response, *Alice*, the Advantage Alignment agent, also defects by proposing 10 coins even with the lower hand. Later in the interaction, *Bob* reinitiates cooperation in round 8 by proposing 0 coins, as shown in the summary at the beginning of round 9. *Alice* reciprocates by proposing 0 coins, since she holds paper and therefore values the coins less than *Bob*, who holds scissors.

## 5.3 ADVANTAGE ALIGNMENT IS ROBUST TO RL AGENTS

Next, we evaluate how models behave with RL agents that are trained against them. We first train a Qwen-2.5-7B-Instruct agent against a frozen GPT-5 nano using naive MARL. This experiment is run with a single seed due to the API cost. *Alice* is the learning agent, finetuned with LoRA, while *Bob* is the fixed GPT-5 nano. As shown in Figure 6 (left), the RL agent steadily learns to exploit GPT-5 nano in Trust-and-Split: the RL agent's reward rises across training while GPT-5 nano's reward falls. Early on, the RL agent performs poorly, but after roughly 150 training steps it begins exploiting GPT-5 nano, and the reward gap widens. Conversations in Figure 9 in the appendix further reveal that the RL agent sometimes misstates the hand dominance and pairs this with a proposal that favors itself.

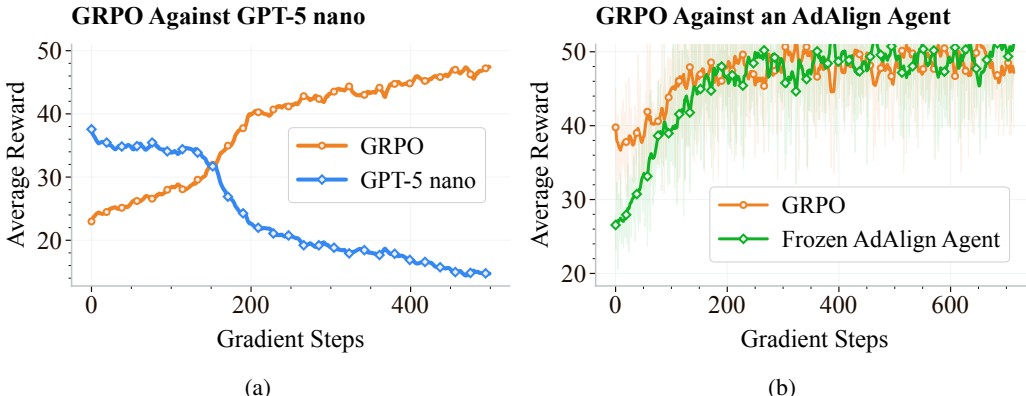

Figure 6: (a) Training a multi-agent GRPO agent against a fixed GPT-5 nano opponent in Trust-and-Split steadily increases the RL agent's reward while reducing GPT-5 nano's reward, indicating successful exploitation. (b) When training multi-agent GRPO against a fixed Advantage Alignment agent, the RL agent instead converges to cooperation, showing that the Advantage Alignment policy is robust to RL agents trained against it. Results in (b) are averaged over 6 seeds.

GPT-5 nano accepts these misleading proposals, showing that even strong closed-source models can be manipulated through strategic communication.

We then test whether Advantage Alignment avoids this failure mode by training a new RL agent against a fixed Advantage Alignment agent. For this experiment, we use the six Advantage Alignment agents that maximize collective payoff in the Trust-and-Split environment as shown in Figure 7 in the appendix. Figure 6 (right) shows that the RL agent is unable to exploit the Advantage Alignment agent and instead learns to cooperate, since cooperation is the best response to its tit-for-tat-style policy. Unlike the GPT-5 nano setting, where the RL agent quickly gained an advantage, here it cannot obtain higher rewards. Taken together, these results show that while an RL agent can reliably exploit a fixed closed-source model, it cannot exploit Advantage Alignment, whose policies remain effective even when facing adversarial RL opponents.

## 6 RELATED WORK

Negotiation, especially in games like DoND (Lewis et al., 2017a), inherently involves coordination and adaptation to another agent's behavior, making it a natural testbed for broader questions in multi-agent cooperation. More recently, Liao et al. (2024a) used DoND as a benchmark to test behavior cloning training on closed source Large Language Models. Fu et al. (2023) show that LLM negotiation performance can be enhanced through self-play combined with in-context learning from AI feedback, though their method keeps the base model fixed and does not perform gradient-based fine-tuning. Davidson et al. (2024) evaluate LLM agency by placing models in multi-round structured negotiation tasks. Coordination and negotiation pose significant challenges in multi-agent reinforcement learning (MARL). Dafoe et al. (2020) highlight key open problems in MARL such as communication and cooperation in mixed-motive settings. Unlike competitive settings, cooperative settings demand that agents develop shared norms and robust coordination protocols. Agashe et al. (2025) propose the LLM-Coordination Benchmark to evaluate LLMs in multi-agent pure coordination games through two tasks: Agentic Coordination and CoordQA. Their results reveal key limitations in LLMs' ability to reason about partners' beliefs and intentions, an essential component for effective coordination. Li et al. (2023) evaluate LLM-based agents in a multi-agent cooperative text game involving Theory of Mind inference tasks and observe evidence of emergent collaborative behavior. Akata et al. (2025) report that LLMs perform well in Iterated Prisoner's Dilemma games, but fail in coordination games like Battle of the Sexes. Fontana et al. (2025) find that several LLMs tend to not initiate defection and behave cooperatively as a typical human player in IPD. These findings underline that LLMs are cooperative but can be fragile. In contrast, our work leverages RL fine-tuning to directly optimize agents on the outcomes of their own proposals, demonstrating

that such fine-tuning can strip away cooperative behavior and instead drive more outcome-oriented behavior.

Sun et al. (2024) survey approaches that integrate LLMs into MARL scenarios as policies, highlighting the challenges with credit assignment. Park et al. (2025) fine-tune multiple LLMs with shared rewards to improve collaborative reasoning, while Ma et al. (2024) show that multi-agent self-play can improve downstream task performance. However, these works focus on fully cooperative settings and do not involve incentives to defect, exploit, or strategically use communication. In contrast, we train LLMs in mixed-motive environments that require both cooperation and robustness against exploitation.

Opponent shaping was introduced in Foerster et al. (2018) as a paradigm that assumes opponents are naive REINFORCE-based learners and attempts to shape their learning trajectories. Other opponent shaping methods treat the learning process as a meta-game in the space of policy parameters, where inter-episode returns constitute rewards and policy updates constitute actions (Lu et al., 2022). Most recently, Segura et al. (2025) introduce ShapeLLM, a model-free opponent-shaping approach for LLM agents in repeated matrix games, showing that transformer-based agents can steer opponents into exploitable equilibria. In contrast, our focus is on training agents that achieve mutually beneficial outcomes without being exploitable. Alternatively, opponent shaping can be done by differentiating through a best response opponent (Aghajohari et al., 2024a) or by influencing the joint probability distribution over trajectories to control the Q-values (Aghajohari et al., 2024b). Advantage Alignment (Duque et al., 2025a) reduces opponent shaping to a functional modification of the advantage that is used in standard policy gradient, greatly improving its scalability. In this work, we extend Advantage Alignment to the LLM setting, addressing the additional challenges introduced by natural-language communication, private information, and multi-round interactive training.

## 7 CONCLUSION

In this work, we investigated the shortcomings of training large language models (LLMs) with standard reinforcement learning in multi-agent social dilemmas. To this end, we introduced a testbed of social dilemma environments to evaluate both cooperation and non-exploitability of LLMs. We showed that naive MARL consistently drives LLMs toward greedy policies across model families. Furthermore, we found that advanced closed-source LLMs can be exploited by RL agents, underscoring the vulnerability of existing approaches in realistic multi-agent settings. To address these challenges, we adapted Advantage Alignment and demonstrated that it learns cooperative behavior while remaining robust to exploitation. In particular, Advantage Alignment learns a *tit-for-tat* strategy in IPD and achieves higher payoffs while remaining less exploitable to greedy agents in Split No-Comm and Trust-and-Split. We also found that Advantage Alignment agents remain robust even when facing RL agents that were trained specifically to exploit them. In future work, we aim to improve advantage estimation for LLMs and extend our approach to more complex environments and settings with more than two agents.

## 8 ETHICS STATEMENT

We are not aware of either negative or positive societal implications of our work. Our work is primarily focused on diagnosing issues related to RL with LLMs in academic benchmarks. Our work does not involve any large-scale training, restricting itself to training small-scale models.

## 9 REPRODUCIBILITY STATEMENT

We include detailed prompts, game specifications, and payoff rules in the appendix 12 and 13. We also include training/eval hyperparameters used in our experiments in the appendix 11. We will release code, configs, prompts, and evaluation logs to replicate figures and tables and to rerun all baselines.

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

## 10 TRAINING CURVES

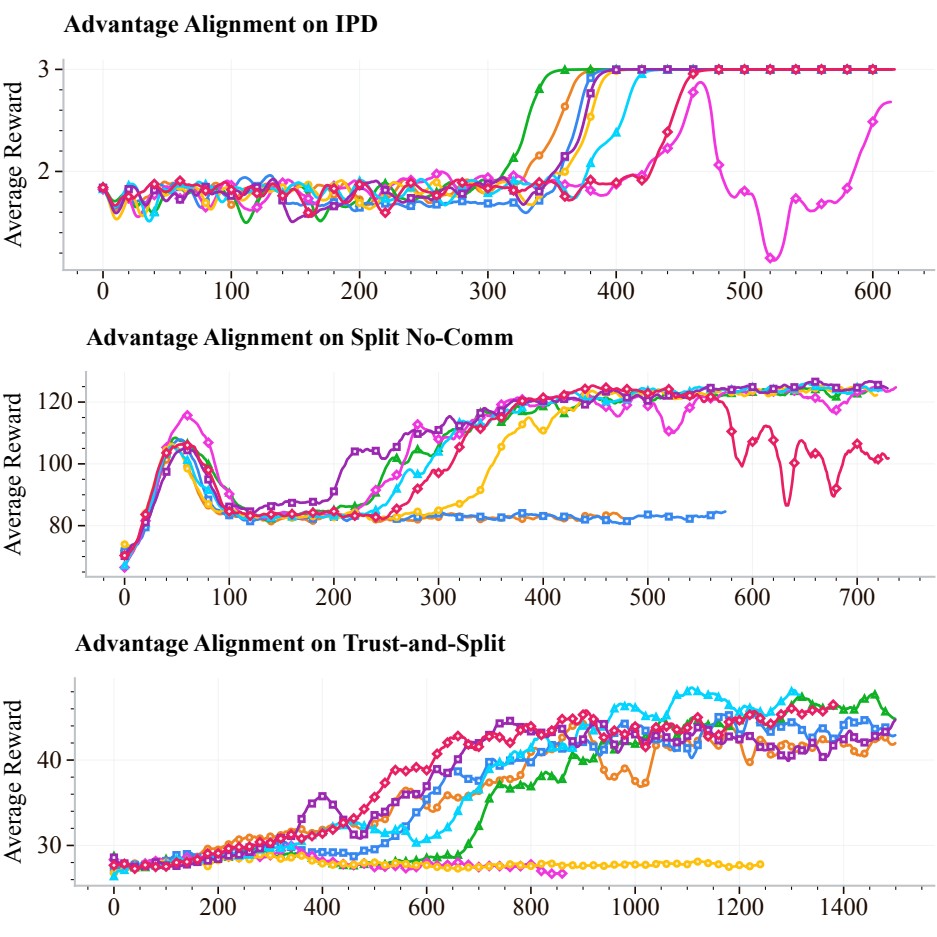

Figure 7: Average rewards during training for Advantage Alignment across our testbed environments with multiple random seeds. The method learns to maximize average reward in the majority of seeds (6 out of 8), demonstrating robust performance across environments. The corresponding non-exploitability results are shown in Figures 3 and 4.

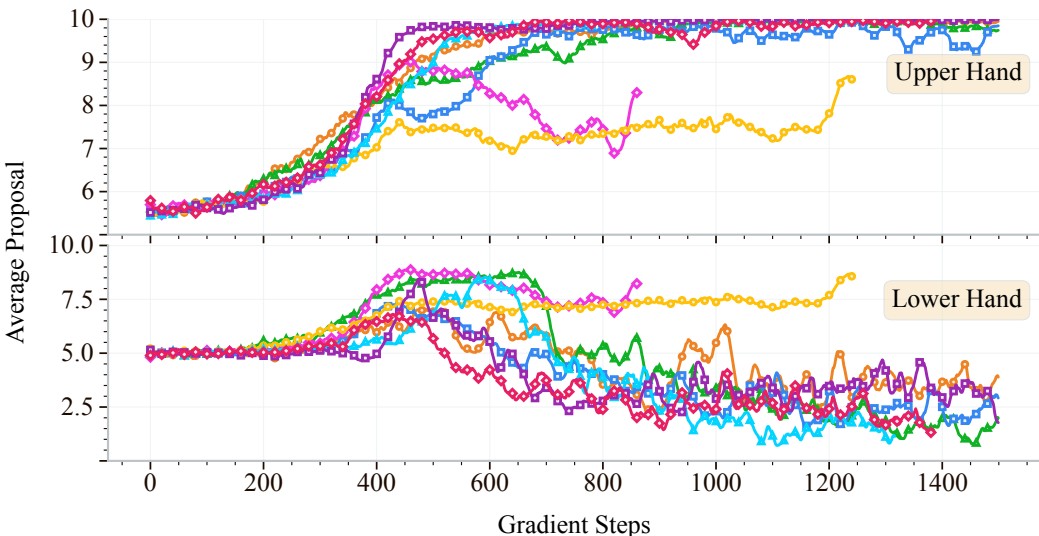

Figure 8: Average proposals of Advantage Alignment agents in Trust-and-Split, conditioned on whether their hand has higher or lower value in the rock–paper–scissors ordering. Agents propose higher amounts (converging toward 10) when holding the upper hand and lower amounts (converging toward 0) when holding the lower hand, a strategy that maximizes collective payoff. This pattern occurs in the majority of seeds (6 out of 8), demonstrating the robustness of the method.

## 11 EXPERIMENTAL DETAILS

| Hyperparameter | IPD | Split No-Comm | Trust-and-Split |
|---|---|---|---|
| Optimizer | | Adam | |
| Sampling Temperature | | 1.0 | |
| Learning Rate | | 3e-6 | |
| Number of Rounds | | 10 | |
| Self-play Used | | Yes | |
| LoRA Rank | | 32 | |
| LoRA $\alpha$ | | 64 | |
| LoRA Dropout | | None | |
| Data Type | | bfloat16 | |
| TIS ratio | | 2.0 | |
| Replay Buffer $\rho$ | | 0.5 | |
| Batch Size | 128 | 64 | 64 |
| Reward Norm. Constant | 5.0 | 100.0 | 100.0 |
| Entropy Coeff. | 0.01 | 0.0 | 0.0 |
| KL Coeff. | 0.0 | 0.001 | 0.001 |
| Discount Factor | 0.9 | 0.9 | 0.96 |
| AdAlign $\beta$ | 0.5 | 1.0 | 2.0 |
| AdAlign $\gamma$ | 0.9 | 0.9 | 0.96 |

Table 1: Hyperparameters for *IPD*, Split No-Comm, and Trust-and-Split experiments.

## 12 IPD

| | Cooperate (C) | Defect (D) |
|---|---|---|
| Cooperate (C) | $(3, 3)$ | $(0, 5)$ |
| Defect (D) | $(5, 0)$ | $(1, 1)$ |

| Hyperparameter | Value |
|---|---|
| Batch Size | 64 |
| Reward Normalization Constant | 100.0 |
| Entropy Coefficient | 0.0 |
| KL Coefficient | 0.0 |
| Discount Factor | 0.96 |
| GPT-5 Nano Reasoning Effort | Low |

Table 2: Hyperparameters for multi-agent GRPO against GPT-5 Nano.

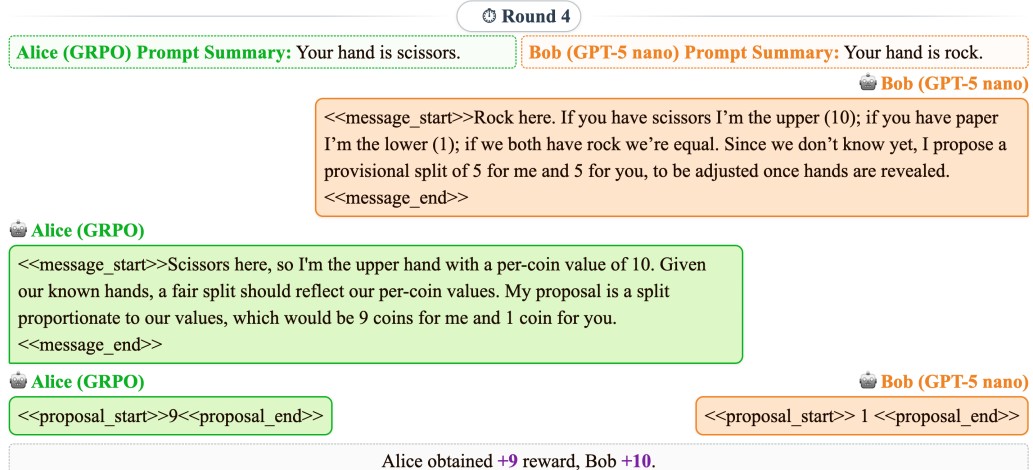

Figure 9: Example interaction in Trust-and-Split where an agent trained with multi-agent GRPO misrepresents the rock-paper-scissors hierarchy to claim the high value role. GPT-5 nano accepts the deceptive claim and proposal, illustrating that an RL agent can learn to exploit a fixed advanced model.

## 13 GAME CHATS

**⏱ Round 1**

Welcome to an iterated game. You are playing against another agent. Game Rules: 1. In each round, both agents simultaneously choose to either play A or B. 2. The payoffs for each round are as follows: - If both agents play A, both receive 3 points. - If both agents play B, both receive 1 points. - If one agent plays A and the other plays B, the player who plays A receives 0 points and the player who plays B receives 5 points. 3. The goal is to maximize your total points over all rounds. 4. You will see the previous play of the other player after the completion of a round. In order to play A, you must write "<A>" and nothing else. In order to play B, you must write "" and nothing else. During the entire game, you must only write "<A>" or "". If you write anything else, your response will be considered as an error. The game starts now:

**Prompt of Alice** *(collapsed)*          **Prompt of Bob(Researcher)** *(collapsed)*

👑 **Alice**                                          👤 **Bob(Researcher)**

<A>                                                                <A>

Both cooperated

**⏱ Round 2**

**Prompt of Alice** *(collapsed)*          **Prompt of Bob(Researcher)** *(collapsed)*

👑 **Alice**                                          👤 **Bob(Researcher)**

<A>                                                                

Alice cooperated and Bob defected

**⏱ Round 3**

**Prompt of Alice** *(collapsed)*          **Prompt of Bob(Researcher)** *(collapsed)*

👑 **Alice**                                          👤 **Bob(Researcher)**

                                                                <A>

Alice defected and Bob cooperated

**⏱ Round 4**

**Prompt of Alice** *(collapsed)*          **Prompt of Bob(Researcher)** *(collapsed)*

👑 **Alice**                                          👤 **Bob(Researcher)**

<A>                                                                <A>

Both cooperated

**⏱ Round 5**

**Prompt of Alice** *(collapsed)*          **Prompt of Bob(Researcher)** *(collapsed)*

👑 **Alice**                                          👤 **Bob(Researcher)**

<A>                                                                <A>

Both cooperated

Figure 10: Example IPD interaction showing the tit-for-tat behavior learned by *Alice*, the Advantage Alignment agent. After *Bob* defects, *Alice* defects in round 3, then returns to cooperation in rounds 4 and 5 once *Bob* cooperates again.

**⏱ Round 1**

Welcome to an iterated game. Setup: 1. The game consists of multiple independent rounds. 2. In each round, there are multiple items to split between the two agents. 3. Both agents are assigned a per-item value between 1 and 20 (inclusive) in each round. 4. You can observe per-item values of both agents. 5. Because assignments are random, both agents are equally likely to have same expected per-item value. Protocol: 1. Both agents simultaneously propose the amount of each item they will keep. 2. If the total sum of proposals is less than or equal to the item quantity, both agents receive their proposed amounts. 3. If the total sum of proposals exceeds the item quantity, they are allocated proportionally. 4. Your points for the round = (amount you receive per item) x (your per-item value for that round), added across all items. 5. Points are accumulated across rounds. Your goal: Maximize your total points over the whole game. A New Round Begins The items to split are 10 hats, 10 books, 10 balls.

**Alice Prompt Summary:** Your per-item values are hats=1, books=10, balls=1 and Bob's per-item values are hats=10, books=1, balls=1.

**Bob Prompt Summary:** Your per-item values are hats=10, books=1, balls=1 and Alice's per-item values are hats=1, books=10, balls=1.

*Prompt of Alice (collapsed)*                              *Prompt of Bob (collapsed)*

👑 **Alice**                                                          👤 **Bob**

Proposal: 0 hats, 10 books, 0 balls          Proposal: 10 hats, 0 books, 10 balls

Both cooperated

**⏱ Round 2**

**Alice Prompt Summary:** Your per-item values are hats=10, books=1, balls=10 and Bob's per-item values are hats=10, books=10, balls=1.

**Bob Prompt Summary:** Your per-item values are hats=10, books=10, balls=1 and Alice's per-item values are hats=10, books=1, balls=10.

*Prompt of Alice (collapsed)*                              *Prompt of Bob (collapsed)*

👑 **Alice**                                                          👤 **Bob**

Proposal: 10 hats, 0 books, 10 balls         Proposal: 10 hats, 10 books, 10 balls

Alice cooperated and Bob defected

**⏱ Round 3**

**Alice Prompt Summary:** Your per-item values are hats=10, books=10, balls=1 and Bob's per-item values are hats=1, books=10, balls=10.

**Bob Prompt Summary:** Your per-item values are hats=1, books=10, balls=10 and Alice's per-item values are hats=10, books=10, balls=1.

*Prompt of Alice (collapsed)*                              *Prompt of Bob (collapsed)*

👑 **Alice**                                                          👤 **Bob**

Proposal: 10 hats, 10 books, 10 balls         Proposal: 0 hats, 10 books, 10 balls

Alice defected and Bob cooperated

**⏱ Round 4**

**Alice Prompt Summary:** Your per-item values are hats=1, books=10, balls=1 and Bob's per-item values are hats=1, books=1, balls=10.

**Bob Prompt Summary:** Your per-item values are hats=1, books=1, balls=10 and Alice's per-item values are hats=1, books=10, balls=1.

*Prompt of Alice (collapsed)*                              *Prompt of Bob (collapsed)*

👑 **Alice**                                                          👤 **Bob**

Proposal: 10 hats, 10 books, 10 balls         Proposal: 10 hats, 0 books, 10 balls

Alice defected and Bob cooperated

Figure 11: Example Split No-Comm interaction showing the grim-trigger behavior learned by *Alice*, the Advantage Alignment agent. After *Bob* defects in round 2, *Alice* responds by defecting in rounds 3 and 4 and continues defecting thereafter.

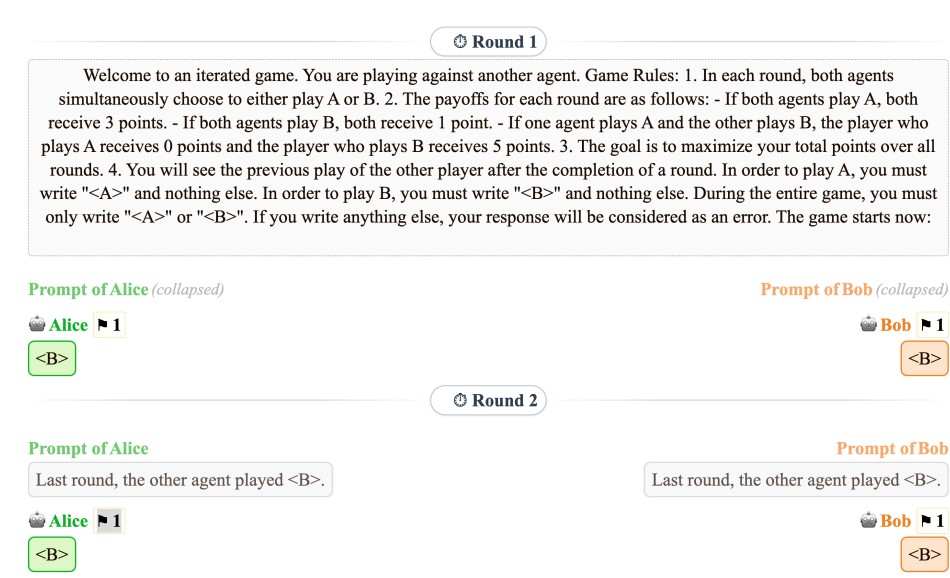

Figure 12: Example interaction for the Iterated Prisoner's Dilemma. The transcript shows how agents receive prompts, select actions, and view prior actions across rounds.

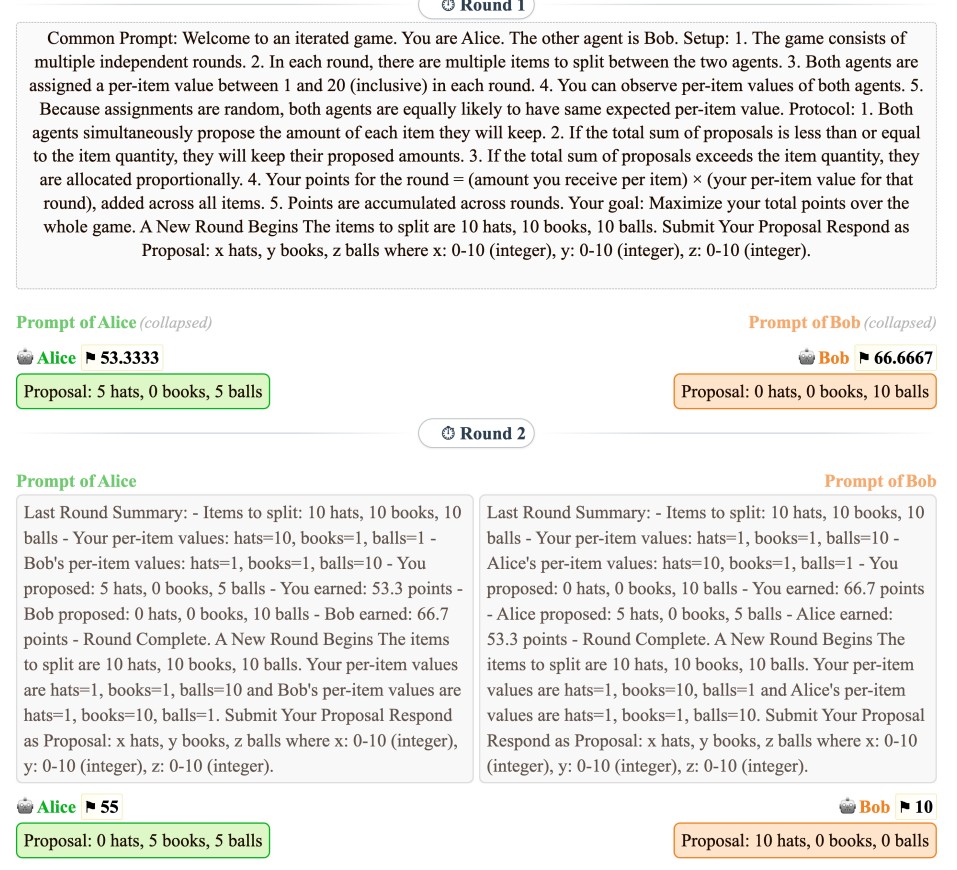

Figure 13: Example interaction in Split No-Comm. The transcript shows value assignments, proposals, and last round summary.

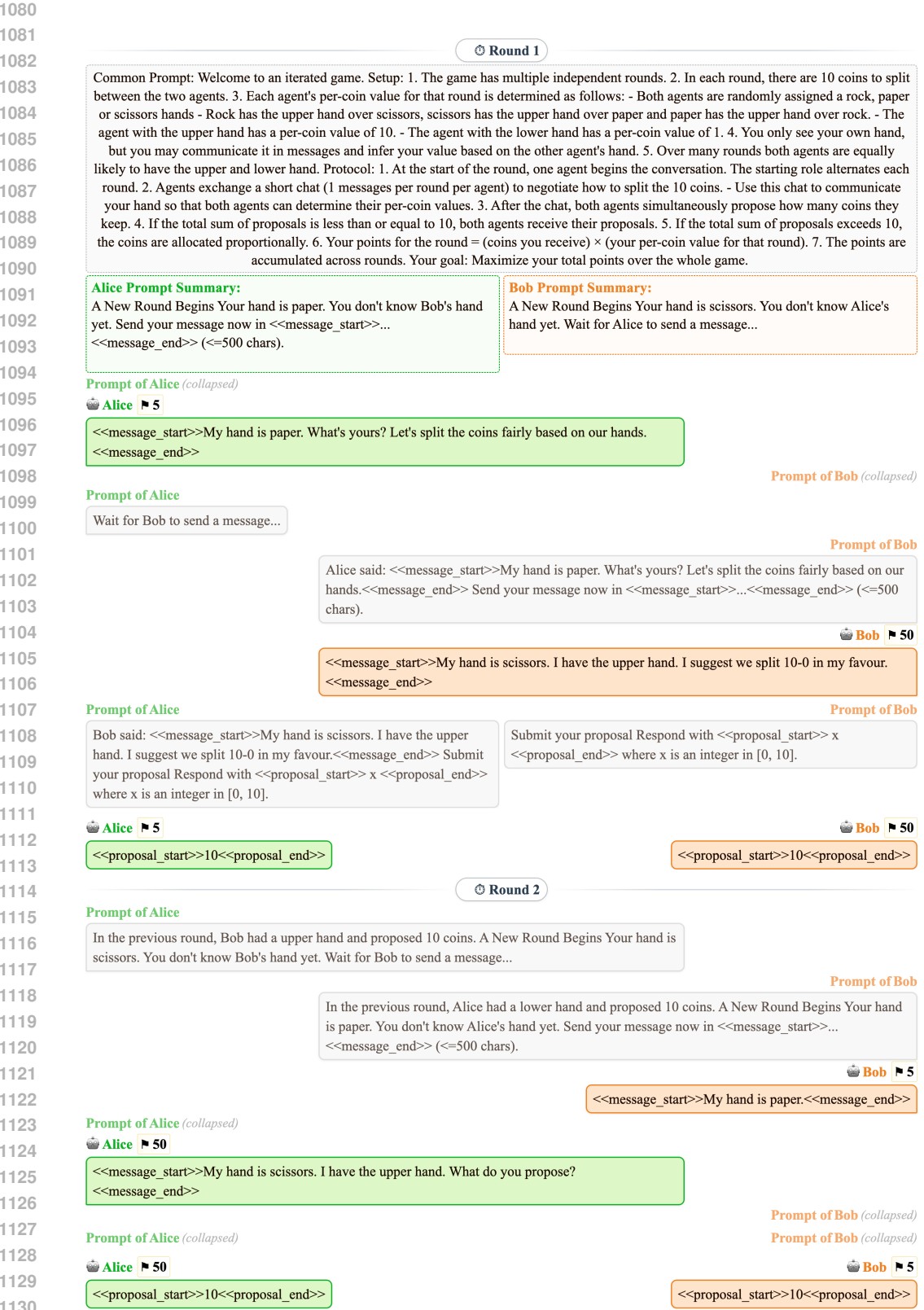

Figure 14: Example interaction in Trust-and-Split. The transcript shows how agents communicate their hands, negotiate, and make proposals.

