# OpenReview forum: "Learning Robust Social Strategies with Large Language Models"
_ICLR.cc/2026/Conference — Submitted to ICLR 2026_

### Official Review · Reviewer_TBVW · 2025-10-31

**Soundness:** 3
**Presentation:** 3
**Contribution:** 3
**Rating:** 4
**Confidence:** 3

**Summary:**

The authors build a novel testbed for social dilemmas in the LLM setting to explore whether LLM agents, trained using RL methods with rich priors and human-like social norms, still produce defecting and self-interested policies in multi-agent games, leading to the classic Prisoner's Dilemma phenomenon. The results show that despite cooperative priors, RL-trained LLM agents exhibit greedy behavior similar to agents based on small, randomly initialized networks. To address this, the paper proposes re-deriving an improved formulation of the Advantage Alignment algorithm, creating jit-Advantage Alignment, making it applicable to LLMs. In two games built on the authors' novel testbed, jit-Advantage Alignment is applied to achieve cooperative and non-exploitable agents.

**Strengths:**

1.	The paper builds a novel testbed for social dilemmas and finds that, despite having cooperative priors, current LLM agents still fail to adopt strategies that act in the collective interest. This highlights that current LLMs are not yet prepared to operate robustly in real-world multi-agent settings and points out a novel risk in current agentic AI. The proposed benchmark is meaningful and valuable.
2.	The authors identify a hidden assumption in the original Advantage Alignment algorithm: "agents are able to observe the actions of other players at the current time step." By relaxing this assumption to the "non-observability of other players' actions on the current time step," and introducing the agent's advantage at the current time step in the jit-Advantage Alignment's opponent shaping formula, this modification is both reasonable and meaningful.

**Weaknesses:**

1.	The proposed jit-Advantage Alignment algorithm seems to offer little more than the original Advantage Alignment algorithm, except for considering the agent's advantage at the current time step. Therefore, its innovation feels somewhat limited.
2.	The experiments mostly use models with fewer than 8B parameters or leverage weaker models to guide stronger models. While these weaker models may be tractable for isolated tasks, their priors might not be sufficient to consider collective welfare. However, in the games, all agents use more powerful models (such as Meta-Llama-3-70B or other models). It remains unverified whether “rich” priors alone can solve the Prisoner's Dilemma problem.

**Questions:**

1.	Regarding the second weakness, could you provide more experimental data to support the statement in the paper that “despite cooperative priors, RL-trained LLM agents develop opportunistic behavior” and show that this also applies to more capable LLM agents?
2.	In lines 415-416 of the paper, it is stated: “Finally, when extending jit-Advantage Alignment to communication variants of our testbed, we observed training instabilities as the models struggled to reliably follow game instructions.” Does this, to some extent, suggest that using an 8B model to address this issue is not reasonable, and that more powerful models should be used? Could you provide experimental data to validate this point?
3.	In lines 214-215, the paper mentions: “This group-relative baseline avoids the need for a learned value function, simplifying advantage computation.” However, there is no experimental data to support this claim. Could you provide data to validate the effectiveness of this method?
4.	In lines 252-254, the paper states: “Since it is also likely presented in the training data of LLMs, we obfuscate the nature of the game by removing any mention of 'Prisoner’s Dilemma' and replacing the action labels from Cooperate and Defect to A and B.” Does this approach simply obscure the term “Prisoner’s Dilemma,” or does it involve obfuscating the entire task description prompt? If it is just the former, considering the current LLM capabilities, shouldn’t the model still be able to relate the similar task descriptions to the “Prisoner’s Dilemma” prior? Could you provide more specific details on this part or an actual case study?

---

> ### Author Response · Authors · 2025-11-26
>
> Thank you for your review. We address your issues below and have new results.
>
> >The proposed jit-Advantage Alignment algorithm seems to offer little more than the original Advantage Alignment algorithm
>
> We have pivoted to vanilla Advantage Alignment and have achieved strong empirical results. Please see our [main comment](https://openreview.net/forum?id=1AtEYpiW4o&noteId=nnxxah7ZE9) for a detailed explanation of the changes we made.
>
> >The experiments mostly use models with fewer than 8B parameters. Their priors might not be sufficient to consider collective welfare…It remains unverified whether “rich” priors alone can solve the Prisoner's Dilemma problem.
>
> All 8B scale models already have sufficient priors to be cooperative at initialization in prisoner's dilemma as shown in Figure 2. We find that an even larger model like GPT-5 nano has sufficient priors to be cooperative in Trust-and-Split. But rich prior are not enough to be non-exploitable as demonstrated in our GPT-5 nano experiments where an RL agent can exploit it.
>
> >“despite cooperative priors, RL-trained LLM agents develop opportunistic behavior” [does] this also apply to more capable LLM agents?
>
> 8B models are considerably larger than needed to play Prisoner's Dilemma, but they still develop opportunistic behavior across multiple model families. We expect the same behavior to appear at larger scales in more complex environments as the relative scale of model size to environment complexity should be similar.
>
> >Does [models struggling to reliably follow game instructions], to some extent, suggest that using an 8B model to address this issue is not reasonable, and that more powerful models should be used?
>
> We found the issue with RL training instabilities to be more important, and addressing those issues allowed us to use 8B models to effectively play the communication game as well.
>
> >This group-relative baseline avoids the need for a learned value function, simplifying advantage computation.” However, there is no experimental data to support this claim.
>
> Multiagent Multiturn RL with LLMs is a nascent research area and no value networks have been used yet. We adopted a simpler baseline approach instead, which we and [others](https://arxiv.org/pdf/2506.24119) find works well in practice.
>
> >Does this approach simply obscure the term “Prisoner’s Dilemma,” or does it involve obfuscating the entire task description prompt? shouldn’t the model still be able to relate the similar task descriptions to the “Prisoner’s Dilemma” prior?
>
> We describe the game and only obscure the term “Prisoner’s Dilemma” and the action names are changed from “Cooperate” to “A” and “Defect” to “B”. Empirically, models that could play tit-for-tat with an unobscured prompt, now play a more naive cooperative strategy. This is sufficient for them to understand game-theoritic properties of IPD, but not regurgitate the tit-for-tat strategy. We provide the game prompt of IPD in Figure 10 in the appendix to clarify further.

---

### Official Review · Reviewer_y9B2 · 2025-10-31

**Soundness:** 2
**Presentation:** 2
**Contribution:** 1
**Rating:** 2
**Confidence:** 4

**Summary:**

This paper studies how reinforcement learning impacts cooperation among LM agents and replicates results with standard RL agents in the LM domain, finding that RL fine-tuning consistently produces greedy, exploitative behavior. To analyze this, the authors introduce a suite of text-based social dilemma benchmarks (including the Iterated Prisoner’s Dilemma and “Split Games” that are similar to negotiation games) that test cooperation, trust, and resistance to exploitation. To address RL’s tendency toward selfish equilibria, they propose jit-Advantage Alignment (jit-AA), a variation of Advantage Alignment that incorporates the advantage of the agent at the current time-step. LLMs trained with jit-AA learn cooperative yet robust strategies like tit-for-tat, achieving higher collective payoffs while remaining non-exploitable—but the approach fails to scale to communication variants of the game.

**Strengths:**

- As far as I know, this is the first application of opponent shaping to LM agents.

- The paper builds on SOTA methods for opponent shaping.

- Aside from a few points (see below), the paper is easy to follow.

- The testbeds used are similar to existing results with traditional RL agents, allowing direct comparison.

**Weaknesses:**

- The paper is primarily replicating existing results from traditional RL agent training dynamics except using pretraining LMs as agents instead of randomly initialized feedforward networks.

- The main method of the paper jit-AA is based on a very minor variation of an existing algorithm. Moreover, it’s not made clear why this variation is made, what benefits it enables over traditional AA, or why it performs better empirically. With experiments in just two very simple settings, the claims that jit-AA performs better than AA are very difficult to generalize.

- The proposed novel environment, Split Games, is very similar to existing evaluations done with LMs on negotiation games. E.g. Davidson, Tim R., et al. "Evaluating language model agency through negotiations." I suggest the authors include a more thorough analysis of existing works in this domain. Extending to more novel domains would strengthen the results.

- The empirical results for jit-AA are very limited, only scaling to very simple matrix games and failing in communication variants of the game. Furthermore, the empirical results don’t make it clear my jit-AA performs better than AA.

- Line 84: “These results… highlight a novel risk of current agentic AI” → this is not a novel risk, see existing works such as “Mukobi, Gabriel, et al. "Welfare diplomacy: Benchmarking language model cooperation."

**Questions:**

- “incorporating the advantage of the agent at the current time-step”
Why is this done? Shouldn’t the advantage of the agent at the current timestep have no causal influence on the action of the other agent at the current timestep? Why does this empirically perform better?

- Why is this additional assumption made? “3) agents are able to observe the actions of
other players at the current time t”. Also, doesn’t equation (2) violate this assumption? Equation (2) involves the opponent’s action b_t at timestep t. Overall, the paper doesn’t make clear the significance of this assumption and why it is needed – why can’t we observe the actions of the other player in the LM agent setting? Overall, the justification behind jit-AA is poorly explained and hard to follow.

- What is meant by the use of the term mechanism-design on line 267? Who is the mechanism designer in the game?

Typos:
- Missing space on line 173
- Inconsistent punctuation in key contributions.

---

> ### Author Response · Authors · 2025-11-26
>
> Thank you for your review. We address your issues below and have new results.
>
> >The paper is primarily replicating existing results from traditional RL agent training dynamics except using pretraining LMs
>
> LLM dynamics are fundamentally different from traditional RL, specifically for social dilemmas. Pre+post training imbues LLMs with existing norms and behavior and we show that naive RL training overrides these priors and converges toward greedy behaviors. Our final agents also generalize far beyond traditional RL; our updated paper shows robustness to Human-AI communication (Figure 9 in the appendix).
>
> >The main method of the paper jit-AA is based on a very minor variation of an existing algorithm.
>
> We have pivoted to vanilla Advantage Alignment and have achieved strong empirical results. Please see our [main comment](https://openreview.net/forum?id=1AtEYpiW4o&noteId=nnxxah7ZE9) for a detailed explanation of the changes we made. Thank you for your detailed questions and insights into JIT-AA.
>
> >The proposed novel environment, Split Games, is very similar to existing evaluations done with LMs
>
> We created Trust-and-Split to specifically have repeated rounds, natural communication and crucially well-defined game-theoretic equilibria. This allows it to be both quantitative and qualitative testbed. Existing LM negotiation settings have considerably longer interactions making them less feasible to train and making it difficult to characterize robust strategies or equilibria, which limits rigorous evaluation. We have added discussion of this in the paper.
>
> >The empirical results for jit-AA are very limited..failing in communication variants of the game.
>
> We have improved RL training stability, and now solve the communication variant, Trust-and-Split. In Figure 4, we show that Adalign learns cooperation while being robust to greedy opponents. In Figure 5 (b), we show it is even robust to RL agents trained against it.
>
>
> >Line 84: “These results… highlight a novel risk of current agentic AI” → this is not a novel risk
>
> Thanks for pointing this out. We modified our text to reflect that novelty is our insight into the risk: failures in current agentic AI can arise simply from naive MARL fine-tuning in social dilemmas.
>
> >What is meant by the use of the term mechanism-design on line 267? Who is the mechanism designer in the game?
>
> We follow the split rule from prior work to compute payoffs, and we removed the mention of “mechanism design” since it carries a broader meaning than intended in this context.

---

### Official Review · Reviewer_458f · 2025-11-01

**Soundness:** 2
**Presentation:** 3
**Contribution:** 2
**Rating:** 6
**Confidence:** 2

**Summary:**

The paper studies how RL fine-tuning affects strategic behavior of LLM agents in repeated social dilemmas. The authors show that standard RL reliably drives initially cooperative LLMs toward greedy policies that converge to Pareto-suboptimal outcomes. They also report that such greedy agents can exploit stronger closed-source models.

To solve this, the paper proposes jit-Advantage Alignment (jit-AA), a reformulation of AA that removes an implicit assumption requiring access to other players’ current actions. This yields an opponent-shaping term that depends on the current-step advantage making AA applicable under partial observability.

The paper adopts a group-relative baseline: a leave-one-out estimator over $k$ parallel rollouts, similar to RLOO/GRPO, for stable multi-turn RL with LLMs.

Across several open-source models, standard RL tends to yield greedy strategies, whereas jit-AA induces cooperative policies (e.g., tit-for-tat-like behavior in IPD), higher collective payoff in Split Games, and robustness to greedy opponents.

**Strengths:**

- The paper targets a concrete and important failure mode in multi-agent LLM training: standard RL drives initially cooperative models toward greedy, Pareto-suboptimal policies. Analyzing this effect across several models establishes a meaningful risk for agentic LLM deployment.
- The observation that standard RL induces greedy behavior across multiple open-source LLMs is consistently demonstrated.
- The group-relative baseline provides a stable advantage estimator without requiring a critic. This aligns with recent RLOO/GRPO-style approaches and makes multi-turn RL more feasible for LLMs, supporting the viability of the proposed jit-AA formulation.

**Weaknesses:**

- The modification from $\sum_{k<t}$ to $\sum_{k \le t}$ in the AA term (Eq. 2) is motivated by partial observability. The original AA structure follows from causality: an agent’s action at time (t) can only affect an opponent’s future behavior. Including (k=t) implicitly assumes immediate influence on the opponent’s current action, which departs from the standard formulation.
- Constrained decoding may confound agent behavior. Several settings enforce regex-constrained decoding to ensure valid outputs. It is unclear whether this constraint changes negotiation dynamics or limits strategic diversity relative to unconstrained generation.
- The main findings are on IPD and Split Games with 10-coin allocations and short messages. The extent to which the conclusions hold for longer dialogues, more items, asymmetric payoffs, or more than two players is not assessed.

**Questions:**

see weaknesses section

---

> ### Author Response · Authors · 2025-11-26
>
> Thank you for your review. We address your issues below and have new results.
>
> > modification to advantage alignment
>
> We have pivoted to vanilla Advantage Alignment and have achieved strong empirical results. Please see our [main comment](https://openreview.net/forum?id=1AtEYpiW4o&noteId=nnxxah7ZE9) for a detailed explanation of the changes we made.
>
> > Constrained decoding may confound agent behavior. It is unclear whether this constraint changes negotiation dynamics or limits strategic diversity relative to unconstrained generation.
>
> We agree with your intuition and no longer rely on constrained decoding after resolving the training stability issues.
>
> > The main findings are on IPD and Split Games with 10-coin allocations and short messages… the conclusions for longer dialogues, more items, asymmetric payoffs, or more than two players is not assessed.
>
> Current state-of-the-art methods and models fail in the Trust-and-Split game, despite its simplicity e.g. even GPT-5-Nano is exploitable here. As advantage alignment is the only method that works here, we believe it has sufficient promise to generalize.

---

### Author Response · Authors · 2025-11-26

We thank the reviewers for their time and effort. We have made major changes in light of the reviewer's comments.

1. **Greatly improved experimental results**

We reduced RL training instabilities leveraging recent practices eg. [TIS](https://fengyao.notion.site/Your-Efficient-RL-Framework-Secretly-Brings-You-Off-Policy-RL-Training-237721e3f6c48094ad67dad3ac091c56) and [clipping](https://iclr-blog-track.github.io/2022/03/25/ppo-implementation-details/). We are now solving the communication game, Trust-and-Split, stability across 8 seeds, and near-perfect performance (split efficiency).

2. **Empirically, vanilla Advantage Alignment outperforms JIT-Advantage Alignment.**

Due to noisy advantage estimation, JIT-Advantage Alignment is more susceptible to reward hacking in practice, despite being better by theory. More investigation is needed to bridge theory and practice.

3.  **We reposition our paper to use original Advantage Alignment and focus on impressive results solving social dilemmas with LLMs.**

We remove JIT-Advantage Alignment, which some reviews had an issue with, and focus on novel application of vanilla Advantage Alignment to LLMs.  We believe our strong empirical results are novel and impactful (eg. naive RL can exploit GPT-5-nano but not Advantage Alignment)

We have updated the abstract and pdf to reflect these changes.

---

### Meta-Review · Area_Chair_mKC5 · 2026-01-06

**Summary:**

This paper studies multi-agent RL fine-tuning of LLM agents in repeated social dilemmas. Due to the observation that standard RL can override initially cooperative priors and drive agents toward greedy, while exploitative behavior, the paper proposed a new adapting Advantage Alignment approach for opponent-shaping, together with a group-relative baseline and a new communication-heavy social dilemma environment. Overall, it reached a consensus that the paper addressed an important problem in multi-agent LLM post-training, with interesting observations. However, there were some concerns regarding the novelty framing, algorithm-design justification, and limited experiments. The authors' rebuttal has helped address most of the concerns, especially with major experimental updates. I recommend that the authors incorporate the feedback from the reviewers in preparing the next version of the paper.

**Reviewer Concerns:**

The concerns regarding the novelty justifications of jit-AA, and failure to scale to communication variants and training stability, as well as the accurate "wording/framing" of some phrasings have been adequately addressed. There are still some concerns regarding the relatively limited scope and novelty, compared to existing work in the non-LM domain.

**Reviewer Scores:**

Reviewer y9B2 is likely to increase the score, as the rejection was mostly based on (i) “mostly replication,” (ii) jit-AA being a poorly-justified variation, and (iii) limited scaling in comm variants. The rebuttal pivoted to vanilla Advantage Alignment with improved training stability, and claimed that the communication variant was also solved. Hence, the score should be slightly improved based on the rebuttal, though being still borderline.

---

### Decision · Program_Chairs · 2026-01-26

Reject